# Successful Lifetime/Long-Term Medical Treatment of Acid Hypersecretion in Zollinger-Ellison Syndrome (ZES): Myth or Fact? Insights from an Analysis of Results of NIH Long-Term Prospective Studies of ZES

**DOI:** 10.3390/cancers15051377

**Published:** 2023-02-21

**Authors:** Tetsuhide Ito, Irene Ramos-Alvarez, Robert T. Jensen

**Affiliations:** 1Neuroendocrine Tumor Centre, Fukuoka Sanno Hospital, International University of Health and Welfare, 3-6-45 Momochihama, Sawara-Ku, Fukuoka 814-0001, Japan; 2Digestive Diseases Branch, NIDDK, NIH, Bethesda, MD 20892-1804, USA

**Keywords:** gastrinoma, Zollinger-Ellison syndrome, PPI, omeprazole, acid hypersecretion, neuroendocrine tumor

## Abstract

**Simple Summary:**

Zollinger-Ellison syndrome (ZES) has, since its original description, been characterized by extreme acid hypersecretion due to a neuroendocrine tumor ectopically secreting gastrin, resulting in severe, recalcitrant peptic ulcer disease/gastroesophageal reflux disease (GERD) that is refractory to standard anti-acid treatments. From the very beginning of its description in 1955, it has been well recognized that control of the acid hypersecretion, both acutely and long-term, is essential to all aspects of management of these patients. Originally, no medical treatment was effective at controlling the acid hypersecretion long-term, resulting in total gastrectomy being the initial treatment of choice. However, starting in the late 1970s with the discovery of histamine H_2_-receptor antagonists (H_2_Rs) and then in the 1980s with the widespread use of gastric H+K+ ATPase inhibitors (also called proton pump inhibitors [PPIs]), medical control of gastric acid hypersecretion became possible in almost all patients. Even though acute acid control studies, as well as short-term maintenance studies (<5 years), suggest that long-term/lifelong acid antisecretory control may be possible, this approach has been called into question recently in both case reports and small series and is becoming increasingly controversial, with the result that the best long-term strategy for treating these patients is becoming unclear. This is occurring because the long-term cure rate, even with increasingly sensitive tumor localization techniques, is <20%; thus, the majority of patients require lifelong treatment. Unfortunately, in contrast to short-term studies, there are no long-term/lifetime treatment studies of acid antisecretory control in ZES patients to address this issue. Whereas studies of long-term/lifetime treatment of patients with GERD are providing insights into the increasing number of questions about the possible long-term side effects of lifelong PPI treatment, which has applicability to ZES patients, no studies are dealing with the larger question about the continued efficacy of medical acid treatment in these patients, as well as the ability to individually manage acid secretion in all patients, which requires a different treatment approach from GERD and thus cannot be addressed by lifelong studies on GERD. The current study attempts to address these issues by analyzing the pharmacology and long-term/lifelong efficacy of medical acid antisecretory agents in ZES patients in a large, long-term NIH prospective study on ZES.

**Abstract:**

Analysis of the efficacy/pharmacology of long-term/lifetime medical treatment of acid hypersecretion in a large cohort of ZES patients in a prospective study. This study includes the results from all 303 patients with established ZES who were prospectively followed and received acid antisecretory treatment with either H_2_Rs or PPIs, with antisecretory doses individually titrated by the results of regular gastric acid testing. The study includes patients treated for short-term periods (<5 yrs), patients treated long-term (>5 yrs), and patients with lifetime treatment (30%) followed for up to 48 years (mean 14 yrs). Long-term/lifelong acid antisecretory treatment with H_2_Rs/PPIs can be successfully carried out in all patients with both uncomplicated and complicated ZES (i.e., with MEN1/ZES, previous Billroth 2, severe GERD). This is only possible if drug doses are individually set by assessing acid secretory control to establish proven criteria, with regular reassessments and readjustments. Frequent dose changes both upward and downward are needed, as well as regulation of the dosing frequency, and there is a primary reliance on the use of PPIs. Prognostic factors predicting patients with PPI dose changes are identified, which need to be studied prospectively to develop a useful predictive algorithm that could be clinically useful for tailored long-term/lifetime therapy in these patients.

## 1. Introduction

Zollinger-Ellison syndrome (ZES) was characterized in its first description in 1955 [1] by a triad of findings, including the presence of severe gastric acid hypersecretion due to a non-beta islet cell tumor of the pancreas, which resulted in refractory peptic ulcer disease/gastroesophageal reflux disease (GERD) that only responded to total gastrectomy [2,3,4,5]. Later studies demonstrated that the acid hypersecretion was secondary to the ectopic release of gastrin by a neuroendocrine tumor (i.e., gastrinoma) [6,7,8], the gastrinomas were more frequently located in the duodenum than in the pancreas [4,9,10,11,12,13,14], and that 20–25% of all patients had ZES as part of the autosomal dominant disorder, Multiple Endocrine Neoplasia-type 1 (MEN1), which is characterized by multiple endocrine tumors/hyperplasia of the pancreas and the parathyroid and pituitary glands [15,16,17,18,19].

From the very beginning, it became apparent that the major cause of the acute- and long-term morbidity and mortality in these patients was not the gastrinomas, which were malignant in a majority of cases and generally slow-growing [20,21,22], but instead the presence of uncontrolled gastric acid hypersecretion [1,2,4,23,24,25,26,27,28]. Acid secretory studies showed that the untreated basal gastric acid hypersecretory rate averaged 4-fold higher than the upper limit of normal in these patients and could reach almost 10-fold higher than the normal upper limit of basal acid secretion in some patients [29]. Furthermore, the tumor induced-hypergastrinemia caused a stimulatory effect on the cells of the gastric mucosa [30,31,32,33,34], resulting in a 4–6 fold increase in parietal cell mass [20,30,35,36], which secretes gastric acid, and was manifested by a marked increase in maximal acid output capacity [20,37]. This degree of acid hypersecretion in almost all patients could not be controlled by the limited medical therapies initially available (antacids, anticholinergics) and it was not until the late 1970s/early 1980’s, with the introduction of increasingly potent histamine H_2_-receptor antagonists (metiamide, cimetidine, ranitidine, famotidine, nizatidine, etc.), that a possible medical alternative to routine total gastrectomy became possible to control the acid hypersecretion [38,39,40,41,42,43,44,45,46,47,48,49,50,51,52,53,54,55,56,57]. Subsequently, even more potent and longer-acting acid antisecretory drugs, gastric H+K+ ATPase inhibitors (also called proton pump inhibitors [PPIs], i.e., omeprazole, lansoprazole, esomeprazole, pantoprazole, rabeprazole), have become available and are now considered the antisecretory drugs of choice for ZES [14,43,58,59,60,61,62,63,64,65,66,67,68,69,70,71,72,73].

There are several important current controversies in the treatment of ZES, but one central issue to the management of all ZES patients is that the acid hypersecretion needs to be controlled at all times and it is unclear whether this is feasible long-term (>5 years). This uncertainty occurs for a number of reasons. First, only 5–20% of ZES patients are surgically curable by complete resection of the gastrinoma [9,43,74,75,76]. This is because of the frequent multiplicity of the tumors in MEN1/ZES patients [77,78,79,80,81]; the fact that gastrinomas may be microscopic in size and not localizable by even the most sensitive imaging techniques [77,82,83,84]; gastrinomas frequently (>50%) metastasize to adjacent lymph nodes and resections are incomplete [9,43,77,81,84,85]; and 20–40% of all ZES patients present with unresectable hepatic metastases [19,20,22,86,87]. Second, there are an increasing number of reports of failure of long-term H_2_R [20,41,88,89,90] maintenance therapy and even difficulty with the ability of long-term PPI treatment to continue to successfully control the acid hypersecretion [81,84,91,92,93,94,95,96,97,98,99,100,101,102,103,104,105,106,107]. The uncertainty is also primarily due to the lack of data from any long-term/ lifetime treatment studies of acid antisecretory control in ZES patients. This is in marked contrast to an abundance of studies of acute acid control and short-term (<5 yrs) control with small numbers of ZES patients [70,71,72,108,109,110,111,112,113,114,115,116,117,118,119]. This lack of information about the long-term efficacy of antisecretory drugs in ZES is particularly disturbing because the unique constant hypersecretory drive of the unresected gastrinoma requires constant inhibition of its effects on acid secretion, which is unique to ZES and can only be addressed by long-term/lifetime study data. Whereas long-term PPI studies in non-ZES patients, especially in patients with advanced idiopathic GERD, are providing evidence for the safety of lifetime PPI treatment [30,120,121,122,123], which is applicable to chronic treatment of ZES patients, unfortunately this is not the case with long-term/lifetime efficacy data in ZES. This is because of the marked variation in the dose requirements between individual ZES patients as well as in each patient, which has been well-examined in short-term ZES acid antisecretory studies [108,109,110,111,112,113,114,115,116,117,118,124].

The current study aimed to address these issues. The NIH prospective study on all aspects of ZES has been in effect since 1974, and part of the study involves the long-term medical management of the acid hypersecretion of all patients [37,74,86,125,126,127]. In the current study, we analyzed data from this prospective database related to all issues of the long-term efficacy of both H_2_Rs and PPIs in these patients. This includes a detailed analysis of initial patient drug dosing for each class of acid antisecretory drug, the final drug dose after long-term/lifetime treatment, including how drug use and dosing has changed with time, as well as the dosing frequency and effects of dosing frequency on total daily dosing. Additionally, we compared acid control with the two classes of antisecretory drugs and finally, we analyzed factors that could be used to predict the need for a possible dose change in long-term PPI-treated patients. Our results demonstrate long-term/lifetime treatment of acid hypersecretion is possible in all patients with each class of acid antisecretory drug; however, PPIs are preferable because of their greater potency and long duration of action. We also comment on a number of aspects of the comparative pharmacology of these two antisecretory drug classes in long-term treatment of ZES patients and provide important guidelines on how similar results can be obtained at other treatment centers.

## 2. Materials and Methods

The patients in this study included a cohort of all patients who have been entered into an ongoing NIH prospective study of various aspects Zollinger-Ellison syndrome (ZES) since 1974, as approved by the clinical research committee of the National Institute of Diabetes, Digestive and Kidney Diseases of the National Institutes of Health, the characteristics of which have been previously described [37,125,126,128,129]. This cohort includes all patients with established ZES who received acid antisecretory treatment with either H_2_Rs or PPIs, with antisecretory doses set by the results of gastric acid testing, as previously described and reviewed briefly below [60,108,112,130,131].

### 2.1. Initial Investigations 

The diagnostic criteria for Zollinger-Ellison syndrome were previously described [14,132,133,134,135,136] and included an elevated fasting serum gastrin level in the presence of an elevated basal acid output, positive provocative testing with secretin or calcium [129,137,138,139,140], a positive histological diagnosis of gastrinoma, or a combination of these criteria.

Basal acid output (BAO) was measured after discontinuing all oral antisecretory medication (greater than 30 h for histamine H_2_-receptor antagonists and greater than 7 days for PPIs) by collecting four consecutive 15-min samples of gastric fluid and titrating them to pH 7.0 with 0.01 N sodium hydroxide [37,59]. Maximal acid output was measured by collecting four 15 min samples following subcutaneous administration of pentagastrin [37,59]. Fasting serum gastrin (FSG) values were determined by Bioscience Laboratories (New York, NY, USA) and all samples were diluted to the normal range for accurate determination of higher values [138]. All gastrin determinations were made during a period when the patient was known to produce gastric acid in order to limit the likelihood of obtaining falsely elevated levels due to drug-induced achlorhydria [14,59,60].

Upper gastrointestinal endoscopy was performed following sedation with meperidine (Demerol; Winthrop Pharmaceuticals, New York, NY, USA) and midazolam (Versed; Roche Laboratories, Nutley, NJ, USA) as previously described [100,108,130]. The upper gastrointestinal mucosa was carefully examined, and biopsies were taken if any abnormalities were detected [31,32,108,141].

Tumor imaging studies included selective abdominal arteriography [142,143], CT scan [144,145], ultrasonography [146], magnetic resonance imagining [147,148], and since 1995, somatostatin receptor imaging [149,150,151,152]. Selected patients with negative imaging underwent functional imaging by assessing gastrin tumor gradients using either portal venous sampling or selective intra-arterial secretin injections [143,153]. Patients with potentially resectable disease underwent surgical exploration, which included particular attention to the duodenum (duodenotomy, palpation, intraoperative transillumination) since 1987 [74,154,155,156,157,158]. Patients with unresectable liver disease (e.g., multiple liver metastases) underwent percutaneous liver biopsy to confirm the nature of their metastatic disease [85,127,159] as well as in patients with distal metastases [43,74,75]. An exploratory laparotomy was not generally performed [151,154]. Patients were placed in the disease-free category post-operatively only if their fasting serum gastrin concentration in the presence of known gastric acid secretion was not elevated, if they had a negative secretin provocative test, if all imaging studies were negative for recurrent tumor, and if there was a significant reduction in symptoms of gastric acid hypersecretion as well as antisecretory drug requirements [29], i.e., they were assessed as having no evidence of ZES [74,127,160]. Patients who were rendered disease-free were generally not treated with PPIs unless they had significant acid-peptic symptoms that were not easily controlled with histamine H_2_-receptor antagonists [161] but responded to PPIs. Surgical patients who were not rendered disease-free (>60%) [74] or who relapsed were treated with PPIs for long-term control of gastric acid hypersecretion.

The possible presence of MEN-1 was considered in all patients with ZES and was evaluated by obtaining a careful family and personal history as well as by performing an endocrine evaluation of the pituitary and parathyroid glands and the pancreas [126,162]. The diagnosis of MEN1 directly affected the management of both the tumor itself and the gastric acid hypersecretion. Specifically, patients with MEN1/ZES are more resistant to acid antisecretory medications if hyperparathyroidism is not controlled [108,163,164]; thus, these patients are usually treated with higher and more frequent drug dosing [108,163,164,165]. Patients with MEN-1 were not routinely subjected to exploratory laparotomy unless they had an imageable tumor (>1.5–2 cm) as studies show the likelihood of rendering them disease -free is low (<5%) without aggressive surgery [14,74,166,167]. Consequently, almost all MEN1 patients required long-term medical control of the gastric acid hypersecretion [14,166].

### 2.2. Determination of the Antisecretory Drug-Dose

Prior to 1983, the acid hypersecretion of all patients was treated with histamine H_2_-receptor antagonists [H_2_Rs] (cimetidine, ranitidine) either alone or with anticholinergic agents [168,169,170,171,172], and afterwards, in almost all cases by either more potent H_2_R’s (famotidine) [112] or PPIs (omeprazole, lansoprazole) in almost all cases [108,110,131,173]. In all cases, the initial and subsequent dosing was determined by acid titration with gastric acid output measurement for the hour prior to the next dose. The usual criteria was to identify the drug dose that reduced the acid output to <10 mEq/h for this time [59,168,174] with an absence of acid-peptic symptoms (<5 mmol/h for patients with severe GERD [130] or prior partial gastrectomy [175]). These criteria were established to induce peptic ulcer healing, control non-severe GERD symptoms, and prevent further mucosal damage [38,108,174]. Initially, H_2_R dosing frequency was every 6–8 h starting with 300/600 of cimetidine, 150/300 mg of ranitidine, or 20/40 mg of famotidine and adjusting the dosage upward or downward as previously described [59,112,169] to reduce the acid hypersecretion to the control criteria above. A similar approach was used for PPIs, initially starting with a once daily dose of 60 mg/day omeprazole/lansoprazole and increasing the dose by 20 mg/day omeprazole or 15 mg/day lansoprazole until a dose of 120 mg/day was reached, and then the daily dose was split into 60 mg twice per day and subsequently increased as needed [108,110,131]. Because initial studies [108,130,175] suggested that patients with complicated ZES (MEN1, previous Billroth 2 surgery, or with advanced GERD) generally required higher dosing, new patients with these features in later studies were initially started on twice per day 40–60 mg/day PPI. After PPIs were licensed for general use in the USA, many new ZES referrals to the NIH were already taking PPIs at the time of their initial evaluation. In these instances, after assessing the BAO to establish the diagnosis off all acid antisecretory drugs, the initial dose of PPI at the NIH was not re-titrated, provided that gastric acid output was effectively controlled as defined above unless this dose induced total achlorhydria. Because studies suggested prolonged drug treatment could induce achlorhydria and result in low vitamin B_12_ levels in these patients [176], the daily referral PPI doses were reduced to levels allowing acid values in the control range without total achlorhydria in uncomplicated cases showing total achlorhydria.

Once the initial acid antisecretory dose requirement was determined for each patient, patients were reevaluated at least once a year to assess both tumor status and confirm that control of gastric acid output remained effective, as previously described [108,109,169,170]. During this reassessment, all patients had a complete history and physical examination, fasting serum gastrin levels and other standard blood biochemical/hematological parameters (i.e., electrolytes, BUN, CBC, etc.) were measured, gastric acid output control was assessed, upper gastrointestinal endoscopy was performed with routine biopsies of the gastric body in the region of the greater curvature, and repeat tumor imaging studies were performed, as previously described [74,75].

During follow-up evaluations, long-term antisecretory drug maintenance doses were titrated upwards in patients with inadequate control or were titrated downwards in patients who were rendered achlorhydric, as previously described [108,109,112,169,170,177]. However, after the completion of a successful formal dose-reduction protocol [108] in a subgroup of the patients, long-term maintenance doses were reduced in amenable patients who were well controlled but not achlorhydric, provided that the criteria for adequate control of gastric acid hypersecretion were not violated.

In the results, to enable comparative drug dose analyses with individual patients treated with different types of H_2_Rs (cimetidine, ranitidine, famotidine) or different PPI’s (omeprazole, lansoprazole) at the time of initial acid antisecretory treatment and final treatment, the doses for H_2_Rs were converted to ranitidine-equivalent doses, as previously described [109,112], or to omeprazole-equivalent doses, as described below. A previous study [112] in ZES patients established that the relative potencies of famotidine: ranitidine: cimetidine are 1: 9:32, with ranitidine being 3.6-fold more potent than cimetidine [112]. To calculate an omeprazole-equivalent dose, results were used from a recent study which demonstrated that 20 mg omeprazole = 40 mg esomeprazole, 30 mg lansoprazole, 40 mg pantoprazole, and 20 mg of rabeprazole [178]. In the analysis, we separated the patients into two different groups based on the length of follow-up. The short-term group was treated for <5 years at NIH, whereas the long-term group was treated for >5 years. Five years was chosen for this division because almost all previously reported ZES patients had only been treated with acid antisecretory agents for <5 years [70,71,72,108,109,110,111,112,113,114,115,116,117,118,119]. We wanted to compare our patients to these previously reported patients, while we know there would be no comparative group for the long-term treatment group. We also wanted to divide the patients into these two groups because we thought the length of follow-up could affect the results, and this allowed us to perform that comparison.

### 2.3. Statistics

Values are expressed as means ± S.E.M. The statistical methods employed for all comparisons were the Fisher’s exact test or Mann Whitney U test. *p*-values of <0.05 were considered statistically significant.

## 3. Results

The clinical, laboratory, and tumor features of the 303 patients included in the current study are listed in Table 1 and Table 2. The results are generally similar to those in other large series of patients with ZES [20,29,86,125,126,179,180,181,182,183,184,185,186] in regard to age at either onset or diagnosis of ZES, gender, percentage with MEN1, presentation of symptoms, disease duration, frequency of prior gastric acid reduction surgery, fasting gastrin levels, basal and maximal acid secretory rates, as well as tumor location and extent. The percentage of patients with duodenal gastrinomas was lower than reported in recent surgical series [9,43,77,81,84,126,187] because the present series included all ZES patients, both medical and surgical, who had their acid hypersecretion controlled in the NIH prospective study. In this prospective study, 28% of ZES patients with advanced metastatic disease (unresectable liver or distant metastases), MEN1/ZES with imaged pancreatic tumors ≤1.5–2 cm, or an accompanying serious medical illness that limited life expectancy did not undergo routine surgical exploration [74,75,127,188], so the exact site of the primary tumor was not established surgically and the imaging studies were inclusive.

**Table 1 cancers-15-01377-t001:** Patient clinical characteristics.

Characteristic	Number (%)
Patient number	
Total treated	303 (100%)
Number followed long-term (≥5 yrs) (a)	260 (86%)
Number not followed long-term (<5 yrs) (a)	43 (14%)
Age at ZES onset (yrs) (b)	
Mean ± SEM	40.0 ± 0.7
(Range)	(11.0–65.8)
Age at ZES Diagnosis (yrs) (c)	
Mean ± SEM	45.8 0.7
(Range)	(11.0–65.8)
Gender	
Male	170 (56%)
Female	133(44%)
MEN1 present	89 (29%)
Presenting clinical symptoms/features (d)	
Pain	233 (77%)
Diarrhea	220 (72%)
GERD (moderate/severe)	133 (44%)
Ulcer history	205 (67%)
Bleeding	70 (23%)
Other GERD/PUD Complication (e)	39 (13%)
Prior gastric acid reduction surgery	37 (12%)
Vagotomy-pyloroplasty/Selective vagotomy	14 (4.6%)
Billroth I resection	6 (2.0%)
Billroth 2 resection	17 (5.6%)
Duration of medical acid treatment started (yrs) (f)	
From disease onset	
Mean ± SEM	3.7 ± 0.3
(Range)	(0–26.1)
Prior to ZES diagnosis (yrs) (g)	
Mean ± SEM	3.5 ± 0.3
(Range)	(0.01–26.0)
From time of diagnosis (yrs)	
Mean ± SEM	1.2 ± 0.3
(Range)	(0–18)

Abbreviations: ZES, Zollinger-Ellison syndrome; MEN1, Multiple Endocrine Neoplasia type 1; yrs, years; GERD, gastro-esophageal reflux disease; PUD, peptic ulcer disease; PMD, patient’s private medical doctor. (a). Patients not followed long-term (i.e., <5 yr.) at NIH because of early death, surgical cure [74,127,160], or returned to PMD after diagnosis, stabilization, and evaluation. (b). Onset defined as onset of recurrent symptoms compatible with ZES, as previously described [125,126]. (c). Criteria for diagnosis of ZES required assessment of BAO/fasting gastrin/secretin tests, as previously described [37,59,129]. (d). Clinical symptoms were determined as previously described [125,174]. (e). Other PUD complications included non-bleeding complications such as stricture [esophageal, duodenal, small intestine], perforation, penetration, obstruction, and advanced Barrett’s esophagus, as previously defined [100,125,130,174,189]. (f). Time to any medical acid treatment included time from onset/diagnosis to start of treatment with histamine H2-receptor antagonists or PPIs. (g). A total of 193 patients were started on medical treatment prior to the established diagnosis of ZES and the remainder were started on treatment at the time of diagnosis or after diagnosis.

**Table 2 cancers-15-01377-t002:** Patient laboratory/tumor characteristics.

Characteristic	Number (%)
Fasting serum gastrin (FSG) (pg/mL)	
Mean ± SEM	4932 ± 1272
(Range)	(72–286,800)
Median	644
Hi FSG (≥644 pg/mL) (≥median)	149 (49%)
BAO (mEq/h) (no gastric surgery) (a)	
Mean ± SEM	43.0 ± 1.5
(Range)	(1.8–159)
BAO (mEq/h) (previous gastric surgery) (b)	
Mean ± SEM	26.3 ± 3.0
(Range)	(2–94)
Hi BAO (≥36.8 mEq/h) (median output of 276 with BAO)	127 (46%)
MAO (mEq/h) (no gastric surgery) (b)	
Mean ± SEM	65.0 ± 2.0
(Range)	(13–159)
MAO (mEq/h) (previous gastric surgery) (b)	
Mean ± SEM	36.9 ± 4.3
(Range)	(9.0–113)
Hi MAO (≥59.7 mEq/h) (median output of 227 with MAO) (b)	112 (49%)
Primary tumor location (c)	
Pancreas	78 (26%)
Duodenum	114 (38%)
Other (d)	36 (12%)
Unknown (d)	86 (28%)
Tumor extent (e,f)	
Primary only	113 (37%)
Primary and lymph node metastases	82 (27%)
Primary and liver metastases	84 (28%)

Abbreviations: BAO, Basal acid output; MAO, Maximal acid output; (a). A total of 276 patients had a BAO (241 had no gastric surgery and 35 had previous gastric acid reduction surgery) determined, as previously described [37,59]. (b). A total of 227 patients had a MAO (202 had no gastric surgery and 25 had previous gastric acid reduction surgery) determined, as previously described [37,160]. (c). Patients with diffuse liver metastases, with MEN1/ZES, or severe co-morbidities did not undergo routine surgical exploration, as previously described [16,77,155,188], and primary location, if not seen on imaging, was not localized. (d). Non-pancreaticoduodenal primary sites occurred in lymph nodes [190,191], hepato-biliary tract [150,192], ovary, jejunum, mesentery, heart, lung, and gastric antrum [43,193,194,195,196,197], as previously described. (e). Tumor extent was determined as described in the Section 2. (f). In 24 patients, it was not possible to determine whether primary only or primary plus lymph nodes were present via imaging and no surgical exploration was performed.

Among the 330 patients included in this study, 260 (86%) were followed at NIH in the prospective study and the gastric acid hypersecretion was controlled for more than 5 years (long-term follow-up group), whereas the remaining 43 (14%) patients were followed at NIH for ≤5 years (short-term follow-up group) (Table 1). The short-term follow-up group included early death of the patient after entering the study, early surgical cure of the patient [74,127,160,161], or the patient requested a return to their private medical doctor after the diagnosis had been established, the disease was stabilized, and/or the initial treatment plan was defined. No patient in either the short-term or long-term treatment groups treated with PPIs or second-generation H_2_Rs (ranitidine, famotidine) ended treatment because of drug side effects, even with some patients receiving very high daily doses. In our early studies in the 1980s when only cimetidine was initially available, some patients treated with very high doses of cimetidine developed anti-androgen side effects (impotence, gynecomastia) and were switched to ranitidine as soon as it was available. This has been previously reported [171].

Initially, all ZES patients entering the NIH prospective study after 1978 had their gastric acid hypersecretion treated with H_2_R’s (first-cimetidine, later ranitidine) (Figure 1), which were the only effective, generally available, approved gastric acid antisecretory drugs able to control acid hypersecretion in ZES patients at that time [39,169,170,171,198]. This continued until 1983, when other gastric antisecretory drugs became available and were used in the NIH patients, with the PPI omeprazole first becoming available [110,199], followed by the more potent H_2_R, famotidine, in 1987 [112,200], and a second PPI, lansoprazole, becoming available in 1989 [131,201] (Figure 1). The increased potency and longer duration of action of PPIs resulted in them being used for long-term maintenance by >90% of patients in this study by the late 1990s (Figure 1).

In Table 3, the overall schedule for the temporal order of acid drug treatments is summarized for the entire time of treatment, as well as for the duration of various treatments. At some time, the majority (81%) of the 303 patients were treated with an H_2_R; however, this was only for a limited time in most patients because only 13.5% were only treated with an H_2_R (Table 3, Part I).

**Table 3 cancers-15-01377-t003:** Acid treatment: Schedule and Duration.

Characteristic	Number (% Total)
I. Treatment schedule	
Total acid treatment drugs (*n* = 303) (a)	
H_2_R (cimetidine, ranitidine, famotidine, nizatidine) at any time	245 (81%)
Only H_2_R without any PPI treatment at any time	41 (13.5%)
H_2_R with anticholinergeric agent at any time	50 (16.5%)
H_2_R without anticholinergeric agent at any time	195 (64.4%)
H_2_R then PPI (b)	204 (67%)
PPI (omeprazole, lansoprazole, pantoprazole) at any time	262 (86%)
PPI only	58 (19.1%)
PPI then H_2_R (c)	0 (0%)
First medical acid treatment drug (*n* = 303) (a)	
H_2_R	245 (81%)
PPI	58 (19%)
Pts with only short-term acid medical treatment <5 yrs (*n* = 42) (d)	
H_2_R only without any PPI treatment at any time	18 (6.6%)
H_2_R with anticholinergic agent at any time	7(2.3%)
H_2_R without anticholinergic agent at any time	35(11.6%)
PPI only	7 (2.3%)
H_2_-R then PPI	17 (6.9%)
Pts with any long-term acid treatment (≥5 yrs) (*n* = 261) (d)	
H_2_R only without any PPI treatment at any time	22 (7.3%)
H_2_R with anticholinergic agent at any time	19 (6.2%)
H_2_R without anticholinergic agent at any time	93 (30.7%)
PPI only	149 (49%)
H_2_R, then PPI	90(30%)
II. Treatment duration (yrs)	
All acid treatment (*n* = 303)	
Mean ± SEM	13.7 ± 0.5
Range	(0.08–48.1)
Only Short-term acid medical treatment <5 yrs (*n* = 42)	
Mean ± SEM	2.8 ± 0.2
Range	(0.08–4.9)
Any Long-term acid treatment ≥5 yrs (*n* = 261)	15.5 ± 0.5
Mean ± SEM	(5.1–48.1)
Range	
Any treatment with H_2_R (*n* = 245)	
Mean ± SEM	6.4 ± 0.45
Range	(0.07–29.2)
Any treatment with PPI (*n* = 262)	
Mean ± SEM	9.8 ± 0.4
Range	(0.1–48.1)

(a). In the NIH perspective trials, histamine H_2_-receptor antagonists were the first effective acid antisecretory medical therapy starting with cimetidine in in 1978, ranitidine in 1982, and famotidine in 1983. PPIs were first used in 1983 with omeprazole, and then lansoprazole in 1989 (See Figure 1). Thus, all patients initially enrolled in this study were first treated with H2Rs (cimetidine, ranitidine, famotidine) and later, most changed to PPIs (omeprazole, lansoprazole), while new patients generally started treatment with PPPIs [110,112,131,171,173,201,202,203]. (b). Patients with active disease were initially treated with H_2_Rs and then switched to PPIs. (c). Patients with active disease were initially treated with PPIs and then switched to H_2_Rs. (d). For explanation of short- and long-term treatment, see Table 1 footnote (a).

Before the availability of PPIs, almost 1 in 5 patients treated with an H_2_R required the addition of anticholinergic agent to control their acid hypersecretion, which corroborated with previous reports that the addition of an anticholinergic agent can potentiate the acid inhibitory effectiveness of an H_2_R in a small number of ZES patients [186,198,204]. When an H_2_R was used in a patient with active ZES, it was used prior to the use of PPIs in all cases, primarily due to the time span of the availability of these drugs (Figure 1). Similar to treatment with H_2_Rs, most (86%) of the 303 patients were treated with a PPI at some time; however, only a minority (19.1%) of patients were treated only with a PPI during their disease course (Table 3). In general, the acid treatment sequences for the 261 patients who had long-term follow-up (i.e., >5 yrs) were similar to those who underwent only short-term follow-up (<5 yrs), with only 6–7% being only treated with an H_2_R and 2–6% requiring an anticholinergic agent in addition to the H_2_R (Table 3, Part I). However, for the 86% of patients who had long-term follow-up, a higher percentage of patients were treated with only a PPI or had the initial treatment with an H_2_R changed to a PPI (30.7% vs. 2.3% and 30% vs. 6.9%, respectively) (Table 3 Part I).

In contrast to previous acid antisecretory studies with ZES patients, the duration of acid treatment in the patients in this study was long (>2–6.5 yrs) [59,110,111,112,113,114,116,118,169,205], with a maximum follow-up time of 48 years, a mean for all patients, including those treated with short- and long-term follow-up, of 13.7 years, and 33% of patients were treated for longer than 15 years (Table 3, Part II). For the 261 patients in the long-term follow-up group, the mean duration of treatment was 15.5 years with a range of 5.1–48 years (Table 3, Part II). For all patients, the mean duration of treatment with a PPI was more than 50% longer than that with an H_2_R (9.8 vs. 6.4 yrs) (Table 3, Part II). During the prospective study, 99 (33%) patients died, none from acid-related problems; therefore, life-time control of the acid hypersecretion was shown in 33% of all ZES patients.

The initial and final acid antisecretory doses for all patients who received H_2_Rs (*n* = 245) at any time or PPIs (*n* = 262) are compared in Table 4 and Figure 2 and Figure 3.

The initial H_2_R daily ranitidine-equivalent dose for all patients was 946 mg/day, but it showed a wide range from 80 to 4800 mg/day for individual patients (Table 4, Figure 3) and the final dosage, which was a mean of 6.4 years later, was 3.6-fold higher (2440 mg/day) (Table 4). With PPIs, the mean initial dose was 72 mg/day, also with a wide range (20–240 mg/day) (Figure 3); however, in contrast to the rise in final mean dose with H_2_Rs, the final dose of PPIs decreased to 58 mg/day (Table 4). A more detailed analysis showed the pattern of dose changes with H_2_Rs and PPIs varied significantly and in the opposite directions, with an increase in dosage required in 5.6 times more patients with H_2_Rs than with PPIs (*p* < 0.0001), a decrease in dose occurring 22 times more frequently in patients with PPIs than with H_2_Rs (0.0001), and no change in dose occurring 1.5 times more frequently in patients with PPIs than with H_2_Rs (*p* = 0. 0037) (Table 4, Figure 2 and Figure 3). These results were comparable to the distribution of the lowest and highest doses of H_2_Rs or PPIs at the initial and final dosing (Table 4, Figure 3). This analysis showed a marked decrease (i.e., 2.5-fold) in the percentage of patients with the lowest final dose and a marked increase (i.e., 4.6-fold) in the percentage of patients with the highest dose for patients treated with H_2_Rs, whereas the opposite pattern was observed in patients treated with PPIs (i.e., 2.5-fold increase in lowest, 1.3-fold decrease in highest) (Table 4 and Figure 3). Similar results in the total patient group were also observed in both the short- and long-term treatment groups (Table 4, Figure 2 and Figure 3).

Numerous studies have shown that in addition to the dosage of H_2_R or PPI given, the frequency of dosing can also have a marked effect on acid suppression efficacy in both ZES patients and patients with acid peptic disorders and/or GERD [59,108,130,168,169,170,178]. To further assess this possibility in ZES patients, the change in dosing frequency between the initial and final dosage for H_2_Rs and PPIs was analyzed in all patients, as well as those receiving short- and long-term acid suppression by each class of antisecretory drug (Table 5, Figure 4). Overall, in the 241 patients taking H_2_Rs at some time, the initial daily dosing frequency was 2-fold higher per day (2.8 times/day) than that for patients taking PPIs (1.5 times per day) and this difference was further increased at the final dosing (mean interval, 6.4 years for H_2_Rs and 9.8 years for PPIs) to 2.4-fold higher in patients taking H_2_Rs than PPIs (3.5 vs. 1.48 times/day) (Table 5, Figure 4). A patient-by-patient comparison of dosing frequency changes between the initial and final H_2_R and PPI doses demonstrated that an increase in dosing frequency occurred in almost half of the H_2_R-treated patients and that a decrease in dosing frequency was uncommon (i.e., 5.6%) (Table 4, Figure 4). In contrast, an increase in dosing frequency was uncommon in patients treated with PPIs (15%), as was a decrease in PPI dosing frequency (Table 5, Figure 4). Overall, this resulted in an increased dosing frequency occurring in almost 3-fold more patients taking H_2_Rs than in those taking PPIs, a 2.3-fold higher percentage of PPI-treated patients having a decreased dosing frequency, and 1.4-fold higher percentage of PPI-treated patients not showing a change in dosing frequency (i.e., 72% vs. 51%) (Table 5).

In patients with idiopathic peptic ulcer disease/GERD, a number of studies reported that splitting the daily dose of PPI may have greater efficacy than increasing a single dose, and splitting the PPI dose could enable use of a lower total daily PPI dose. Similar conclusions have been proposed from studies of small numbers of patients with ZES [131,173,206,207]. To further investigate the relationship between increasing the drug dosing frequency and its effect on the total daily dose in ZES patients, we compared the effect on daily drug dosage of increasing the dosing frequency in 99 ZES patients treated with H_2_Rs and 39 patients treated with PPIs (Figure 5). In H_2_R-treated patients, an increase in drug dosing frequency almost always (93%) resulted in an increased total daily dose of the drug, which was in marked contrast to what occurred with PPIs (Figure 5). In contrast to H_2_Rs, splitting the daily PPI dose resulted in a decreased dose in 13 of patients and an increase in daily dose of only 46% of patients (Table 4). These results supported previous proposals that splitting the dose will frequently allow control of the acid hypersecretion with a lower daily total PPI dose in ZES patients with unsatisfactory acid control, rather than increasing the daily dose [108,109,131]. In contrast, with H_2_Rs, splitting the daily dose may be necessary to control the acid hypersecretion in ZES patients because of the shorter duration of action of these drugs [88,112,169,200,201], but it will not generally result in a lower total daily dose (Figure 5).

The data showed the percentage of ZES patients treated with either a histamine H_2_-receptor antagonist or proton pump inhibitor at any time, during short-term (<5 yrs) or long-term (≥5 yrs) treatment who required an increase in daily drug frequency, no dosage frequency change, or had a decrease in dosage frequency, determined as described in the Section 2. The durations between the first and last daily doses used for comparing daily dosing frequency were the same as listed for the daily dose comparison in the legend of Figure 2.

Table 6 shows the treatment outcomes with both H_2_Rs and PPIs for acid output and symptom control. In both H_2_R- and PPI-treated patients, the initial mean untreated BAO was almost 4-fold higher than normal [161] (40–42.2 mEq/h), and the mean acid output was reduced to the therapeutic treatment level of <10 mEq/h prior to the next dose of antisecretory drug (or lower for patients with GERD/Billroth 2 surgery as described in the Section 2) in both the initial and final dosing.

For both treatment acid assessments (i.e., initial/final acid control), the mean acid outputs were 2–3-fold higher when H_2_Rs were used than when PPIs were used (4.2/3.3 vs. 2.2/0.97 mEq/h for initial/final acid control, respectively), although both were well within the therapeutic control range (i.e., <10 mEq/h) (Table 6). In both H_2_R- and PPI-treated patients, symptoms (i.e., pain, diarrhea, GERD-symptoms, etc.) due to the acid hypersecretion [20,40,125] were controlled in all patients, all peptic ulcers/esophagitis or other mucosal abnormalities due to the acid hypersecretion [174] healed if initially present, and new lesions were prevented.

At present, in >95% of all ZES patients, gastric acid hypersecretion is controlled by PPIs (Figure 1). Identifying possible predictors of which patients might require daily dose changes with PPI continued treatment could be of major clinical value in allowing more tailored follow-up of acid control in these patients. To attempt to identify such predictive factors, we analyzed the possible predictive value of various clinical, laboratory, and tumoral features and previous acid control values for identifying which patients required a subsequent daily PPI dose change (Table 7).

**Table 7 cancers-15-01377-t007:** Predictive value of various clinical, laboratory, tumor, or acid features for identifying patients requiring at least one dose change during PPI treatment between the first and final doses (a).

Variable	Number (Percent)		
	Change Daily PPI Dose (*n* = 123)	Same PPI Dose (*n* = 139)	*p*-Value
Clinical			
GERD			
Any (b)	77/156 (49%)	53/106 (50%)	0.99
Severe (b)	24/50 (48%)	11/212 (5.2%)	<0.0001
Diarrhea at onset (c)	121/156 (78%)	78/106 (73%)	0.47
PUD/GERD complication (d)	31/156 (65%)	17/106 (19.8%)	0.52
>6.4 yrs from ZES onset to treatment (c)	76/156 (49%)	53/106 (50%)	0.90
Age > 40 yrs at ZES onset (c)	74/156 (47%)	51/106 (48%)	0.99
Male gender	91/156 (58%)	60/106 (57%)	0.80
MEN1 present (c)	43/156 (27.6%)	35/106 (33%)	0.41
Prior gastric acid reduction surgery (d)			
Any acid reduction surgery	18/156 (11.5%)	7/106 (6.6%)	0.21
Prior Billroth2	10/148 (6.3%)	0/104 (0%)	0.007
Lab			
High BAO > 36.8 mEq/h (e)	78/156 (50%)	39/106 (37%)	0.043
High MAO > 62 mEq/h (e)	99/216 (89%)	12/12 (100%)	<0.0001
High FSG > 644 pg/mL (e)	78/156 (50%)	48/106 (45%)	0.53
Tumor features			
Primary tumor size (f)			
≥3 cm	42/156 (26.9%)	34/105 (32.3%)	0.41
<1 cm	98/156 (63%)	60/105 (57%)	0.37
Localized Disease (g)	110/156 (70%)	80/206 (75%)	0.40
Liver Metastases	46/156 (29.5%)	26/106 (24.5%)	0.40
Pancreatic primary	24/108 (53%)	21/67 (31.3%)	0.21
Duodenal primary	64/109 (59%)	33/67 (49%)	0.27
Acid treatment			
Length PPI Tx ≥ 9.5 yrs	84/156 (54%)	46/106 (43%)	0.10
Acid control 1st PPI dose (h)			
0 mEq/h	7/91 (7.5%)	29/144 (32%)	0.0095
≥5–9.9 mEq/h	23/143 (16%)	10/92 (10.9%)	0.34
PPI daily dose (1st PPI dose)			
QD: Daily PPI dose > 55 mg/day (i)	54/80 (68%)	26/80 (32%)	0.0267
BID: Daily PPI dose ≥ 80 mg/day (i)	60/71 (84%)	31/47 (15%)	0.0252
Frequency 1st PPI dose: >1×/day	71/156 (46%)	47/76 (62%)	0.90
H_2_R dose day > 600 mg prior to PPI	59/124 (48%)	35/83 (42%)	0.48

Abbreviations: GERD, gastro-esophageal reflux disease; PUD, peptic ulcer disease; yrs, years; MEN1, Multiple endocrine Neoplasia type 1; OM, omeprazole; ZES, Zollinger-Ellison syndrome; Tx, treatment; 1st PPI dose, initial dose patient treated with PPI determined by acid titration, as described in the Section 2 and previously in ref. [108,109,110,131,156,173]; UGI, upper gastrointestinal; for other abbreviations, see legends for Table 1 and Table 2. (a). The median duration with PPI treatment between first and last dose was 9.8 ± 0.4 years, during which time patients were seen yearly and PPI dosage was checked by assessing acid suppression and adjusted, if needed, as described in the Section 2. (b). GERD was determined by history, UGI endoscopy, and the presence of heartburn and dysphagia; severity was determined as previously described [100,125,130]. (c).Presence of diarrhea, time of ZES onset, presence of MEN1, and length of time from ZES onset to diagnosis/treatment were determined as previously described [37,125,126,162,208]. (d). PUD/GERD complications and previous gastric acid reduction surgery were determined as described in Table 1 legend. (e). High BAO, MAO, or FSG were determined as values exceeding the median values for all patients. (f). Tumor size was determined at surgery or by detailed imaging studies, as previously described [142,144,147,209]. (g). Localized disease was defined as occurring in patients with proven ZES with no evidence of liver or distant metastases via detailed imaging studies or surgery, as detailed in the Section 2. (h). The first PPI acid control was assessed by measuring gastric acid output 1 h prior to the next dose of PPI and is reported in mEq/h, which was determined initially and then later assessed yearly, as described in the Section 2. (i). PPI dose is expressed as omeprazole-equivalent dose, as described in the Section 2.

Significant factors for predicting a PPI dose change were the presence of severe GERD (not mild/moderate GERD); previous history of Billroth 2 surgery (but not other acid reduction surgery, such as vagotomy, pyloroplasty, or Billroth 1 surgery); the presence of a high BAO (i.e., >36.8 mEq/h) or high MAO (i.e., >62 mEq/h); or the presence of achlorhydria on the initial acid control evaluation (Table 7). A number of other factors reported in small numbers of ZES patients [100,108,109,113,130] with possible predictive value for drug dose changes or correlations with higher acid secretory rates were found not to correlate with PPI dose changes. This latter group included the presence of MEN1; presence of acid secretory complications; moderate GERD; large tumor size; more advanced disease; or the presence of higher acid control levels (Table 7).

## 4. Discussion

The primary purpose of this study was to determine whether long-term/lifelong gastric acid antisecretory control was possible and effective in patients with ZES and to study the pharmacology of the antisecretory drugs during this prolonged treatment period. This study was undertaken for several reasons. First, lifelong treatment of gastric acid hypersecretion continues to be required by most ZES patients. This continues to occur because only 5–20% of all ZES patients are cured surgically [9,43,74,75]. This low overall cure rate is due to multiple causes, including the fact that 20–40% of all ZES patients present with unresectable hepatic metastases [13,19,20,22,86,87]; 25% have ZES/MEN1 and a 0–5% long-term cure rate because of the characteristic presence of multiple, metastatic, and small duodenal gastrinomas that cannot be cured without aggressive resection, such as Whipple resection, which is not generally recommended [74,77,80,126,166,188]; and in the remaining 30–40% of ZES patients with possible resectable gastrinomas or with sporadic ZES, the long-term cure rate is only 20–40% [9,43,74]. Second, the lifelong period of maintenance acid treatment eventually needed by most ZES patients is long because the average age of ZES onset is between 27–44 years [43,86,125,210] and the mean survival rate is 90% at 25 years after onset in MEN1/ZES patients and >60% in sporadic ZES patients, thus most patients require acid secretory control for >25 years [19,187,211]. Third, while numerous studies have reported the effectiveness of both different H_2_Rs and PPIs in initially controlling acid hypersecretion in ZES patients [41,68,87,89,113,115,116,118,168,205,212,213,214,215,216], numerous studies have raised questions about the long-term efficacy of histamine H_2_-receptor antagonists (H_2_Rs) to fulfil the need for life-long therapy for these patients because of their lower potency/duration of action, resulting in the requirement for high, frequent doses that increase with time [41,88,89]. Furthermore, more recently, increasing numbers of reports have raised similar concerns about the long-term effectiveness, as well as safety, of PPIs for controlling lifelong acid hypersecretion in these patients [81,84,84,91,92,93,94,95,96,97,98,99,100,102,103,104,105,106,107], even though this class of acid antisecretory drugs is more potent and longer-acting than H_2_Rs in ZES patients [43,131,217]. Fourth, there is minimal data on the pharmacology of long-term (i.e., >5 yrs) H_2_R or PPI treatment in ZES patients. Short-term maintenance studies provide evidence that maintenance dose adjustments may be frequently required in ZES patients treated with H_2_Rs or PPIs, including the need for increased dosing, decreased doses, or no dosing change in some patients; however, the frequency of these changes in long-term treated patients remain unclear. Furthermore, the development of drug tolerance has been proposed in both H_2_R- and PPI-treated ZES patients [88,89,109,113,170], but the possible frequency or severity remain unknown. Some studies, but not others [108,109,113], have suggested that acid antisecretory dose reduction may be possible with prolonged antisecretory treatment in ZES patients, but its frequency or durability remain unclear. In addition, some studies suggest that, pharmacologically, it may be better to alter the antisecretory drug dose interval rather than altering the drug dose at a specific time for optimum antisecretory control [108,109,170]. However, relatively few patients have been studied to fully resolve this issue. Finally, if antisecretory doses need to be altered in some ZES patients with time, this can only be established at present by performing periodic acid secretory testing, which is invasive and needs to be performed in all patients because there is currently only minimal information about who may require a drug dose adjustment at some point. Hence, a systematic study attempting to identify patients who may require a dose adjustment could be of marked clinical importance in patient management by helping to better tailor acid antisecretory control during maintenance.

Unfortunately, with the data available at present, it is not possible to address the issues raised in the previous paragraph about the long-term/lifelong efficacy/pharmacology of acid antisecretory treatment in ZES patients. The primary reason for this conclusion is that there is a lack of studies on long-term/lifelong acid antisecretory treatment in ZES patients. With both different H_2_Rs and PPIs for ZES patients, all maintenance acid antisecretory studies have been short-term (<1–5 yrs), with only small numbers of patients followed for longer durations, thereby limiting the ability to perform detailed analyses or pharmacologic studies [108,109,110,111,112,113,114,115,116,117,118]. The present study has none of these limitations and allows the question of efficacy /pharmacology of long-term/lifelong acid antisecretory treatment in these patients to be systematically addressed. In this study, the data was collected prospectively from a large number of ZES patients (i.e., 303 patients) as part of the NIH prospective study on ZES, alterations in drug dosing/frequency were performed according to a set protocol, both the efficacy of H_2_Rs and PPIs was studied to allow comparisons, the study had a long follow-up providing data for up to 48 years of antisecretory treatment with a mean treatment period of 14 years, and treatment was lifelong for 33% of the patients.

The most important conclusion of the present study is that long-term/lifelong medical treatment of acid hypersecretion in ZES patients is possible and can be successful in controlling peptic acid symptoms and preventing the development of acid-related complications in all patients. While this statement includes long-term/lifelong treatment with either a PPI or H_2_R, for all practical purposes, the findings with PPIs will be of primary interest to most clinicians. Presently, because of the greater potency and longer duration of action of PPIs, which allows once or twice a day dosing, PPIs have become the drugs of choice for acid treatment in ZES [14,43,168,218,219]. PPIs have almost completely replaced the use of H_2_Rs, as shown in this study, with PPI use almost completely replacing of H_2_R use, which is similar to their relative use in other recent studies of acid control in ZES patients [43,115,218,220]. These results differ from several recent case reports and small series [81,84,91,92,93,94,95,96,97,98,99,100,102,103,104,105,106,107] that have suggested long-term treatment with PPIs did not satisfactorily control the gastric acid hypersecretion in some ZES patients. They also differ, in the case of long-term treatment with H_2_Rs, from a number of studies that have reported poor results maintaining acid secretory control in ZES patients [20,41,88,174]. The primary difference between this study and others that have also successfully controlled acid hypersecretion in ZES patients for shorter periods (i.e., 1–6 yrs) [108,109,110,111,112,113,114,115,116,117,118] and those that report a high rate of drug failure in maintenance control of acid hypersecretion, is whether systematic acid secretory testing to establish efficacious criteria was routinely used to appropriately titrate both the initial and all follow-up H_2_R/PPI doses. Numerous acid secretory control studies in patients with ZES support the conclusion that acid antisecretory drug dose requirements for both PPIs and H_2_Rs can vary greatly among patients and thus need to be individually titrated using established criteria for acid suppression in these patients [14,59,89,130,174]. In ZES patients with intact stomachs, without moderate to severe GERD or MEN1, antisecretory drugs inducing acid suppression to <10 mEq/h for the hour prior to the next drug dose is the generally accepted criteria [14,43,59,89,115,116,118,131,174]. In contrast, in patients with complicated ZES (moderate-severe GERD, previous Billroth 2 surgery, or MEN1) greater acid inhibition to <1 mEq/h may be needed depending on the UGI endoscopic findings [100,108,113,130].

Numerous short-term acid antisecretory studies with PPIs [109,110,113,115,118,131,201] or H_2_Rs [109,112,169,170] in ZES patients have reported that not only can the daily total dose vary markedly between individual patients, but it can also change considerably in different directions for different patients over time. In our study, the initial daily dose determined by upward or downward drug-dose titration based on acid secretory results (expressed as omeprazole-equivalent dose) for the 262 patients treated with PPIs was 72 mg/day and the initial daily dose for the 245 patients taking H_2_Rs (expressed as a ranitidine-equivalent dose) was 968 mg/day. However, for both PPIs and H_2_Rs, the daily dose varied markedly for individual patients with a range of 20–240 mg/day for PPIs and 80–4800/mg/day for H_2_Rs. The mean and range of doses for initial treatment for both PPIs [109,110,111,114,117,201,221] and H_2_Rs [109,112,169,170] were similar to those reported in various short-term studies in ZES patients using similar methods to establish initial drug dosing. The marked difference in daily doses required by individual patients (12-fold for PPIs, 60-fold for H_2_Rs) demonstrate why individual dose titration is required to successfully manage the acid hypersecretion in these patients, because at present, the required dose needed by each patient cannot be established by any other method. On the final acid assessment after a mean of 9.8 years (range 0.1–48.1 yrs) for the PPI-treated patients, the mean daily omeprazole-equivalent dose had significantly decreased (*p* < 0.001) by 20% to 58.5 mg/day (range 20–240 mg/day). The direction of the change in the mean PPI dose between the initial and final daily doses was in the opposite direction of the final daily dose seen in patients treated with H_2_Rs, which was significantly (*p* < 0.001) higher (i.e., 2440 mg/day) than that seen with the initial dosing. These results are consistent with short-term studies of ZES patients treated H_2_Rs in which patients generally required increasing H_2_Rs dosing with time, with an average of at least one dose adjustment per year, which was generally an increased dosage [20,59,88,89,90,112,170,222]. These results of long-term PPI treatment were similar to some short-term studies with PPIs in ZES patients [108,223]; however, they differed from other studies generally reporting that increased PPI dosing was needed with time, although it occurred at a much slower rate of 0.13 changes per year than the 1 dose per year generally seen with H_2_Rs in various studies [109,111,113,131,173]. To help resolve this difference between what was previously reported in short-term PPI studies with the PPI results determined in our study, we analyzed the total direction of dose changes for our 303 patients overall and patients undergoing short-term (<5 yrs) or long-term (mean 15.5 yrs, range 5.1–48.1 yrs) treatment. Our results showed there was a large difference between the distribution of overall drug dose changes over time (mean total treatment duration = 13.7 ± 0.5 yrs) between patients treated with PPIs or H_2_Rs. A 6-fold higher percentage of patients treated with H_2_Rs required an increased drug dose (i.e., 70% vs. 12.6%). In contrast, a 22.5-fold higher percentage of PPI-treated patients required a dose reduction compared to patients treated with H_2_R’s (47% vs. 2.1%). Meanwhile, the percentage of patients who did not need a dose change was generally similar in H_2_R- and PPI-treated patients (28% vs. 40%). The comparison of our results from the short- and long-term treatment groups showed that the major difference was a lower percentage of dose reduction in the patients in the short-term treatment group, suggesting this may contribute to the above difference in the frequency of dose changes between previous short-term PPI studies in ZES patients and the results of the long term/lifetime results in this study.

The daily dose change results reported in the above paragraph from our study raise both important pharmacological considerations, as well as important clinical aspects in the long-term treatment of these patients. First, the exact pharmacological basis for the 6-fold higher rate of daily dose increase that is required with H_2_R long-term/lifetime treatment compared to that seen in ZES patients treated with PPIs is unclear. A number of pharmacological studies in patients treated with H_2_Rs, primarily with GERD, report that there is a high rate of developing tachyphylaxis with continued treatment [224,225], whereas most [224,226,227], but not all studies [113], report a lack of tachyphylaxis with continued PPI treatment. In our study, the failure to find any correlation between markers of either tumoral function, aggressiveness, or tumor growth and the increasing H_2_R dose requirement suggests that the increased H_2_R daily dose requirement is not a direct result of tumor growth/activity and is compatible with the possibility of developing H_2_R tachyphylaxis. This difference in the development of tolerance between H_2_Rs and PPIs in ZES patients has the important clinical consequence that once the initial daily dose is adequately set in ZES patients with a PPI, the likelihood of needing early dose escalation is a much less than with an H_2_R. Furthermore, it raises the possibility that if predictors for PPI dose change can be identified, the requirement during follow-up for repeat acid assessment can be much better tailored toward certain patients. Second, the finding that dose reduction was possible with time in almost half of all PPI-treated patients, but in only 2% of H_2_R-treated patients, has important pharmacological and clinical considerations for the treatment of ZES patients.

The entire subject of whether dose reduction of either PPIs or H_2_Rs should be routinely attempted during maintenance treatment in ZES patients is controversial at present. Some short-term studies report that the daily dose of PPIs can be reduced in up to 58–83% [108,109,201] of ZES patients and that patients were safely maintained on these lower doses, whereas another study [113] reported that the daily PPI dose could be reduced in only 27% of patients and concluded that it could not be generally recommended until further studies were performed. There are no prior comparable studies of attempted dose reduction in ZES patients treated long-term with H_2_Rs. Previous pharmacological time-course studies of acid inhibition with different doses of H_2_Rs (cimetidine/ranitidine), demonstrated that their action was maximal at 5–6 h, followed by a rapid rebound of acid secretion [40,88,112,170]. Furthermore, as seen in the present study and reported previously, in contrast to PPIs, H_2_Rs rarely reduce acid secretion to very low levels in ZES patients (i.e., <2 mEq/h) [40,88,112,170]. No prior studies of systematic dose reduction with H_2_Rs have been reported, likely because of the above pharmacological findings with H_2_Rs. It is generally thought that an H_2_R dose reduction would, in most cases, result in inadequate acid control. The current study supports this conclusion in that dose reduction during maintenance treatment in H_2_R-treated patients was rarely (i.e., 2%) successful.

The resolution of the controversy of PPI dose reduction during maintenance therapy with ZES patients remains a pressing issue primarily for three reasons. First, both recent reviews and numerous papers (primarily epidemiological studies), have raised increasing concerns about the possible long-term/lifetime side effects of PPI treatment. These include PPIs being associated with increased bone fractures/bone problems (dental implants); nutritional/drug malabsorption (vitamin B_12_, magnesium, calcium, iron); increased chronic renal disease; lung disease (pneumonia); cardiovascular disease (CVS) (CVS mortality, myocardial infarction, stroke); CNS abnormalities (dementia); increased infections (Clostridia, bacterial infections with liver disease); increased development of carcinoids and other tumors; and interference with metabolism of important therapeutic agents/drugs [30,31,123,176,224,228,229,230]. Although causality or possible mechanisms of action by which PPIs cause these changes are not clear in most cases, the daily dose of PPI could be a contributing factor in some cases [228,230,231,232]. Because of this, in most guidelines for chronic treatment with PPIs in various diseases increasingly recommend using the lowest doses of PPIs, or even stopping PPI treatment for periods of time, and that PPIs only be used for established indications [123,228,230,231]. A number of these potential PPI side effects have been reported in ZES with long-term PPI treatment [31,43,81,84,94,97,107,118,123,168], particularly, decreased nutrient absorption (Mg^2+^, vitamin B_12_), and development of carcinoid tumors and other neoplasms, which have led to difficulties in long-term acid control and even the need for total gastrectomy. Second, because of the recommended manner of establishing the initial individual drug dose requirement by serial upward dose titration for both PPIs and H_2_Rs, subsequent PPI dose reduction becomes a particularly important issue. This occurs because in most cases when patients are first evaluated for diagnosis and tumor localization/extent, their initial maintenance dose requirement is usually established starting with an omeprazole-equivalent dose of 60 mg/day (or 60 mg twice a day in patients with complicated diseases such as severe GERD, Billroth 2 resection, MEN1) [14,59,60,108,109,111,113,114,116,118,130,131,173,177,201,233]. This approach is recommended and generally used because many patients, when initially seen, have acute mucosal disease/marked symptoms that need to be rapidly controlled to avoid the rapid development of life-threatening peptic-ulcer/GERD complications [2,4,210]. The lowest dose of PPI equivalent to omeprazole (20 mg or 20 BID) could be used with subsequent upward titration, but this may delay adequate dosing and disease control because of the mechanism of action of PPIs [130]. This conclusion was supported by one prospective study [130] in ZES patients where this recommended low dose of PPI (20 mg/day omeprazole) was used in 49 patients to determine whether this method satisfactorily identified the initial dose needed to control the acid hypersecretion. Using this method, one third of all patients failed to reach satisfactory acid control during the duration of the study, and the patients that failed at a low dose could not be clearly predicted [130], leading the authors to recommend that this approach should not generally be used. The mechanism of action of PPIs is via inactivation of the final common pathway of acid secretion by directly binding to and inactivating the activated hydrogen-potassium ATPase pump [234] The active drug only has a short plasma half-life (<60 min) and although it irreversibly inactivates the active pumps, its short half-life and subsequent pump biosynthesis result in the final duration of effect of these drugs [234] with the efficacy of PPIs increasing over the first 3–5 days of use [234]. Therefore, the current recommended initial dosing method allows rapid control of acid secretion, but a proportion of these patients who cannot be predicted will be able to subsequently have their acid controlled on lower PPI doses with time, as shown in our study. The third reason for dose reduction, if possible, is monetary. Most insurance companies do not pay the full amount of the drug cost, thus placing a burden on some patients at various times.

In contrast to issues related to changes in the magnitude of the antisecretory drug dose in ZES patients, which has been well-studied in numerous short-term studies and reviewed in the previous paragraphs, the issue of the frequency of antisecretory daily drug dosing and its possible effect on total daily drug dose has not been well-studied in ZES patients. Specifically, the effects of antisecretory drug frequency have neither been systematically nor well-studied in short-term studies of ZES patients, nor has it been studied in any manner in long-term studies. This is an important area of study both for clinical concerns and pharmacological insights. In studies of patients primarily with GERD, it has been reported that the frequency of administration of both H_2_Rs and PPIs can have a marked effect on their therapeutic efficacy, even to the extent that symptoms and esophageal pH can be controlled with greater ease by regulating the frequency of drug administration, rather than only increasing the dose [178,235,236,237,238,239]. Because of their pharmacological shorter duration of action, this type of clinical response would be predicted particularly with H_2_Rs; however, it is also reported in some studies of GERD patients treated with PPIs at different frequencies and dosages [178,236,237,238,239,240]. A number of our results support the conclusion that altering the antisecretory drug dosing frequency of both H_2_Rs and PPIs in ZES patients can achieve similar results as those reported above in GERD patients due to similar drug pharmacological effects on gastric hypersecretion in ZES patients, thus resulting in similar positive clinical results.

Our results showed that the frequency of both the initial drug dosing as well as the frequency of rate of change in daily dosing during long-term maintenance antisecretory therapy varied markedly with long-term treatment in ZES patients and that the frequency varied markedly between ZES patients treated with long-term maintenance H_2_Rs or PPIs. Specifically, we found that the initial daily dose frequency was 2-fold higher with H_2_Rs than with PPIs, which reflected the known greater duration of action of PPIs over H_2_Rs in both peptic/ulcer/GERD and ZES patients [110,112,131,201,217,234]. Overall, in our study, one half of all patients on long-term maintenance with H_2_Rs demonstrated a change in daily dosing frequency (either increase or decrease), whereas with long-term PPI treatment, almost one third of patients required a change in dosing frequency over the treatment period of a mean of 9.8 years (range 0.1–48.1 yrs). Furthermore, the percentages of patients were similar in both the short-term (<5 yrs) (H_2_Rs = 60 pts, PPIs = 128 pts) or the long-term (H_2_Rs = 117 pts, PPIs = 202 pts) treatment groups, demonstrating that this change in drug dosing frequency was not a function of the length of the follow-up. Not only did the overall frequency of the daily antisecretory dose vary between H_2_R- and PPI-treated patients during long-term maintenance, the direction of the dosing frequency changes differed markedly with a 3-fold high percentage of patients showing an increase in dose with H_2_Rs over PPIs, while a 2.5-fold higher percentage of PPI-treated patients showed a decrease in the daily dosing frequency compared to H_2_R-treated patients. Lastly, in a subgroup analysis of 99 patients, we found that increasing the frequency of PPI dosing resulted in a proportion of patients requiring a lower total daily PPI dose, whereas this rarely occurred in H_2_R-treated patients with increased dosing frequency, which resulted in an increased total daily dose in 95% of patients. The above findings from our long-time/lifetime study are consistent with the results from a few short-term studies on ZES [109,147,173,199,241], supporting the conclusion that dividing the dose can be a better strategy for patients requiring a high daily dose of PPIs, rather than further increasing the daily dose, and may allow a lower effective total daily dosage.

As pointed out in the initial paragraph of the discussion and frequently mentioned in various short-term studies of acute initial control and longer maintenance control of acid hypersecretion in ZES patients, the key to successfully managing acid hypersecretion is the careful assessment of gastric acid hypersecretion control to establish suppressive values, which have been shown to allow mucosal healing and prevent addition mucosal damage [14,43,59,89,114,168,242,243]. As shown in this study and others [108,109,110,113,130,147], both the initial and subsequent assessments of acid secretory control in ZES patients show marked dose requirement variability from patient to patient, hence, the approach needs to be individualized for each patient with appropriate titration of their antisecretory dose [14,108,109,110,113,130,147,244,245]. This requires repeated gastric analysis, which is labor intensive, uncomfortable for many patients, and may require travel to one of the few centers that offer such a service. Because, as shown in the present study and other short-term studies [43,113,114,168], gastric acid hypersecretion is currently treated with PPIs in more than 95% of all ZES patients, both acutely and for maintenance, any predictor that could identify patients who will or will not need a PPI drug dose change could be of marked clinical value in allowing patients to be selected for assessment and better tailoring of care. In the present study, we found a number of clinical and laboratory features that significantly correlated with the subsequent need for PPI dose change during long-term treatment extending over a mean of 10 years (range 1–48 yrs). These variables included the presence of severe GERD, but not mild/moderate GERD; the history of a previous Billroth 2 procedure, but not a previous vagotomy with other drainage procedure; the presence of a high initial BAO (i.e., >36.8 mEq/h) or MAO (i.e., >62 mEq/h); the presence of achlorhydria on the initial acid assessment (i.e., acid output = 0 mEq/h); and two variables related to the initial PPI dosing, including a high PPI daily dose for those taking QD PPI dosing (i.e., >55 mg/day) or a high daily dose for those taking BID/TID dosing (i.e., >80 mg day of omeprazole). These prognostic results overlap with some but not all features previously described in short-term studies in ZES patients that correlate with final drug PPI drug doses, the subsequent ability to reduce PPI drug doses, or the subsequent need to increase PPI drug doses either because of continued/new peptic symptoms or the development of acid-peptic related mucosal changes during maintenance PPI treatment, after first setting the initial dose by acid titration [100,108,109,113,114,130,131]. Our results disagree with previous short-term studies that reported no predictive value of the BAO/MAO or fasting serum gastrin level for the need for a higher/lower dose of lansoprazole during maintenance [113,131]. However, our predictive results agreed with previous short-term studies in which PPI maintenance doses did not correlate with tumor extent [131]; patients less frequently had dose changes when they had previously required lower doses of antisecretory drugs [108,173]; increased dosing changes were required for patients with two of the three features of complicated ZES (i.e., advanced GERD, Billroth 2, but not the presence of MEN1) [108,114]; there was a lack of predictive value for most clinical factors, (i.e., age, gender, duration of disease, or symptoms) [113]; there was a lack of effect of the occurrence of prior peptic ulcer complications on subsequent dosing changes [113]; and with other studies that showed increased PPI doses/dose changes occurred more frequently with prior high BAO [113,173,201,205]. One of these predictive features that caused some confusion is the importance of MEN1/ZES. MEN1/ZES occurs in 20–25% of all ZES patients [15,86,126,165] and numerous studies have clearly established the fact that the presence of hyperparathyroidism in these patients (>95% at some point) [15,86,126,165,246] results in increased BAO, MAO, fasting gastrin levels, and increased resistance to antisecretory drug therapy [163,164,165,247,248]. If hyperparathyroidism is recognized and successfully corrected, which can be difficult in some patients [126,162], these changes were reversed post-parathyroidectomy [163,164,165,248]. The problem is that hyperparathyroidism has a high recurrence rate and can be missed, so this can affect the PPI maintenance dose [163,164]. In our patients who were assessed yearly for the recurrence of MEN1 features, recurrences were rapidly detected and corrected, with the result being that we saw no effect of the presence of MEN1 (i.e., hyperparathyroidism) on drug maintenance doses. The result of identifying these predictive factors for PPI daily maintenance dose changes will need to be prospectively studied to determine whether they can be safely applied in order to enable structured tailoring of gastric analysis, thereby more easily managing lifelong acid hypersecretion in these patients.

It could be argued that this study’s results cannot be extrapolated to most clinical situations because it was carried out as part of the NIH prospective study on ZES, which is a different clinical scenario that that seen in many centers. We think that this is not the case for a number of reasons. First, it is recommended by almost all guidelines [12,14,43,154,218] that patients with ZES should be treated in specialty centers where there is expertise in all aspects of this disease and with other neuroendocrine tumors. A number of these centers exist in the US and have this capability. This is best demonstrated by the fact that many of these centers have the capability of carrying out a treatment course such as outlined in this study, including the measurement of gastric acid outputs in response to drug dosing, attested by the numerous, short-term, and successful acid inhibitory studies in the recent literature. These reports include the successful treatment of acid hypersecretion in ZES patients with various acid antisecretory drugs, including esomeprazole [118,221,249], lansoprazole [63,64,67,70,113,114,131,250,251], pantoprazole [72,73,116,213,252,253,254], rabeprazole [69,115,255], and omeprazole [66,68,106,205,244,256,257]. With the availability of PPIs, which are both more potent and longer acting than H_2_Rs, the ability to control acid hypersecretion in these patients has become more manageable in different centers.

While this study investigated the effectiveness of long-term medical control of the lifelong acid hypersecretion that is present in most ZES patients, it is important to remember that all ZES patients have two problems: control of the acid hypersecretion and treatment of gastrinomas, which are malignant in 60–90% of ZES patients. Surgery remains the only possibility of cure and treatment for both of these problems. With the increased sensitivity of imaging modalities, the possibility of rendering a patient disease-free is increasing and thus, systematic reassessment of possible surgical resection in addition to control of the acid hypersecretion need to be performed.

## 5. Conclusions

The key findings of this study are summarized in Table 8. 

The most important overall conclusion of the current study was that long-term/lifetime medical treatment of gastric acid hypersecretion in uncured ZES patients (>70–90% of all cases) was possible in all patients included in this study. These results should be applicable to almost all series of ZES patients in different centers. The study included a large number of patients (*n* = 303), they resembled most smaller series in the literature in all characteristics, and the cohort had all aspects of the disease, including different tumor extents, with or without MEN1, with or without previous gastric acid reduction surgery, with and without different degrees of GERD, all of which have been reported to affect the success of medical acid antisecretory therapy in ZES patients. Furthermore, the present study was perspective, with a fixed protocol, and extended up to 48 years for one patient (mean 15 yrs), allowing it to be carried out for lifetime treatment in 30% of all patients.

The conclusion of this study is that the lack of efficacy of gastric acid antisecretory therapy in ZES patients does not have to be a major factor in deciding the course of treatment (for example, medical vs. surgical) in these patients. The development of increasingly sensitive imaging methods for localizing these tumors (i.e., somatostatin receptor imaging, etc.) is increasing the possibility of surgical cure; therefore, improved understanding of the natural history of patients with different aspects of ZES, especially those with MEN1/ZES, and the development of novel and effective methods for the treatment of ZES patients with advanced disease can now become the main area of attention for therapeutic approaches.

## Figures and Tables

**Figure 1 cancers-15-01377-f001:**
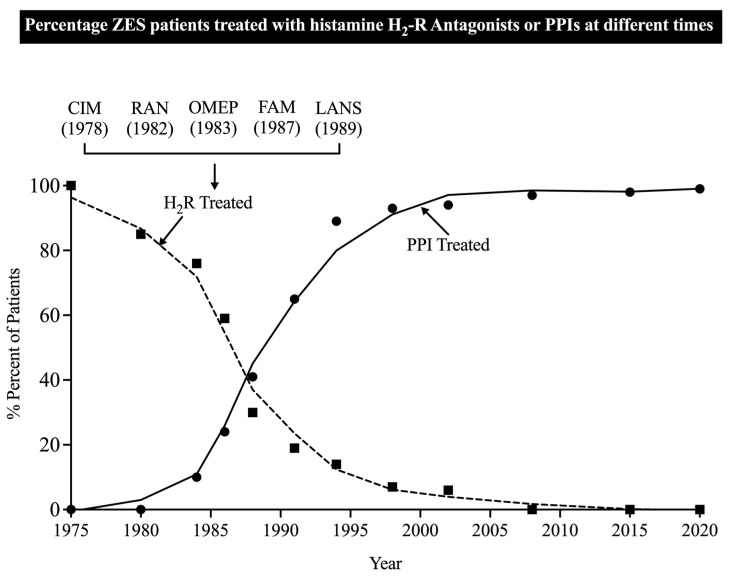
Time course of acid antisecretory drug treatment in 303 patients with ZES. Shown are the percentages of ZES patients with acid hypersecretion during different time periods treated with either histamine H_2_-receptor antagonists (cimetidine, ranitidine, famotidine, nizatidine) or proton pump inhibitors (PPIs) (omeprazole, lansoprazole, pantoprazole). The time periods analyzed were 1/76–1/83 (90 patients), 2/83–1/85 (117 patients), 2/85–1/87 (147 patients), 2/87–1/90 (177 patients), 2/90–1/93 (196 patients), 2/93–1/96 (216 patients), 2/96–1/99 (210 patients), 2/99–1/2002 (188 patients), 2/2002–1/2006 (166 patients), and 2/2006–2020 (91 patients). Data are reported as percentage of patients during the given time period that were treated with either group of drugs.

**Figure 2 cancers-15-01377-f002:**
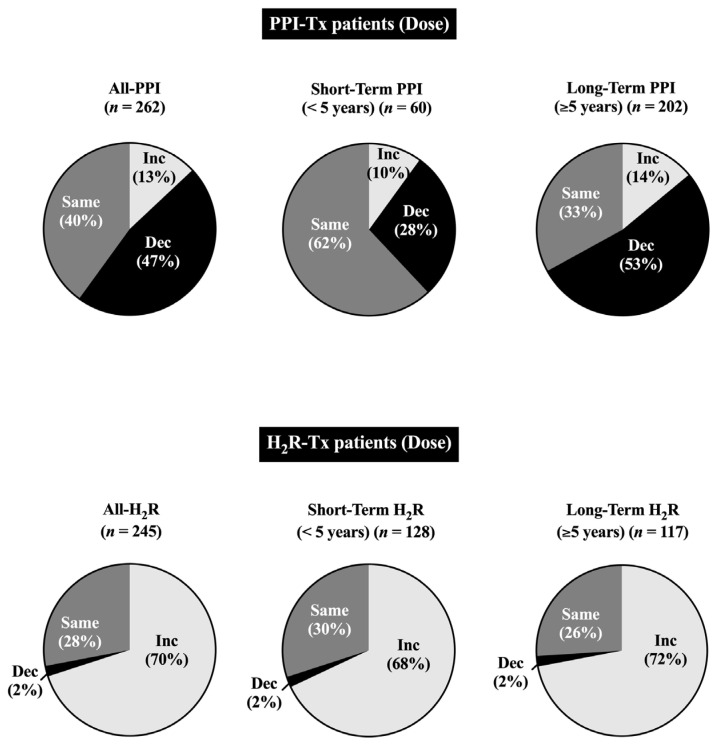
Pie charts showing distribution of changes in drug dosage with short- (<5 yrs) or long-term (≥5 yrs) treatment with either a histamine H_2_-receptor antagonist or proton pump inhibitor. Data shows the percentage of ZES patients treated with either a histamine H_2_-receptor antagonist or proton pump inhibitor at any time, during short-term (<5 yrs) or long-term (≥5 yrs) treatment, who required an increase in daily drug dosage, no dosage change, or a decrease in dosage determined as described in the Section 2. For PPI users, the mean time between the first and last dose for the total was 9.78 ± 0.35 years, the mean time for the short-term treatment group was 2.45 ± 0.17 years, and the mean time for the long-term treatment group was 12.00 ± 0.32 (range 5.2–32.2) years. For H_2_R users, the mean time from first to last dose time for all users was 6.38 ± 0.32 years, the mean time of the short-term treatment group was 2.32 ± 0.13 years, and the mean time of the long-term treatment group was 10.86 ± 0.50 (range 5.0–29.2) years. Abbreviations: Inc.: increased dose, Dec.: decreased dose.

**Figure 3 cancers-15-01377-f003:**
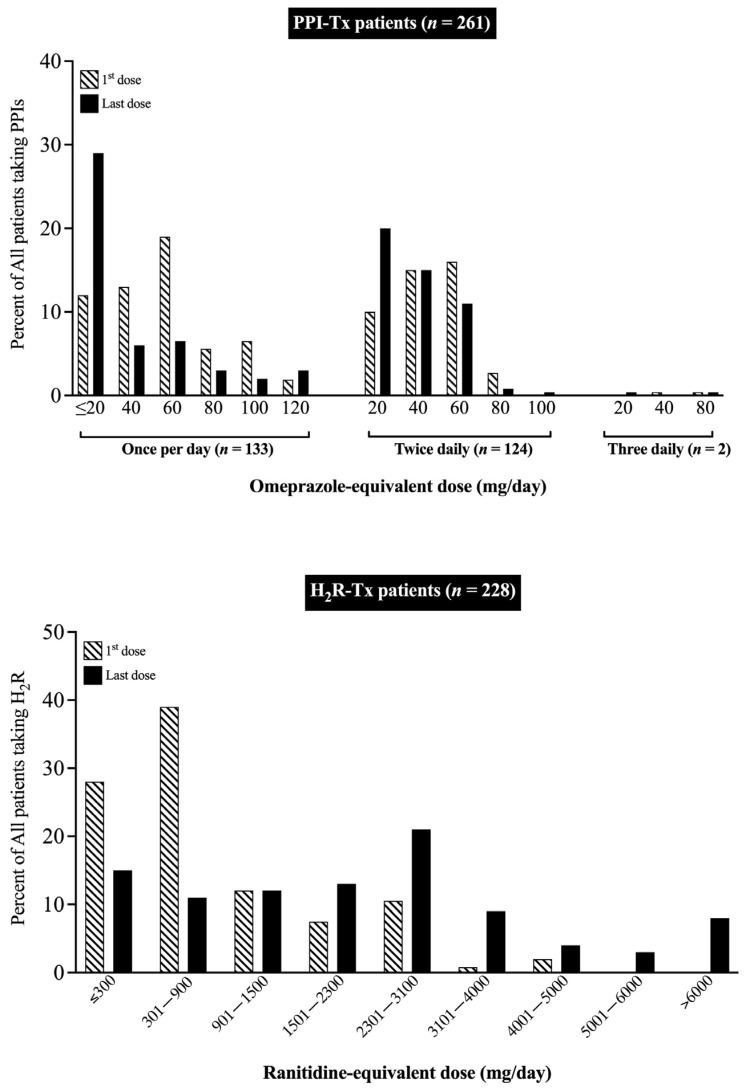
Pie charts showing distribution of changes in drug daily frequency with short- or long-term treatment with either a histamine H_2_-receptor antagonist or proton pump inhibitor.

**Figure 4 cancers-15-01377-f004:**
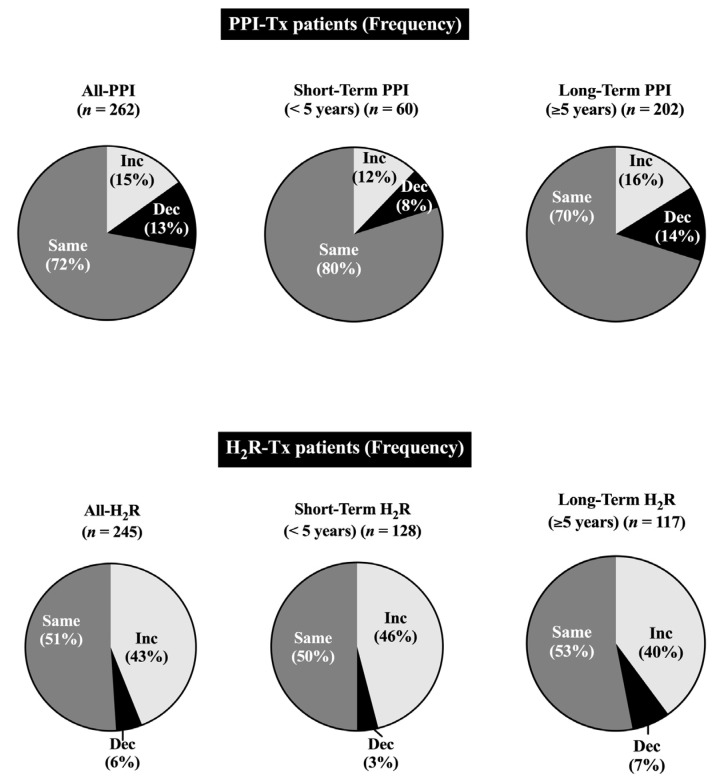
Pie charts showing distribution of changes in total drug daily dose in patients who had an increased frequency of dosing with short- or long-term treatment with either a histamine H_2_-receptor antagonist or proton pump inhibitor. Data shows the percentage of ZES patients having an increased daily dosing frequency treated with either a histamine H_2_-receptor antagonist or proton pump inhibitors at any time, during short-term (<5 yrs) or long-term (≥5 yrs) treatment, which resulted in an increase in total daily drug frequency, no total daily dose change, or had a decrease in total daily drug dose, determined as described in the Section 2. The duration between the first and last daily doses used for comparing daily dosing frequency were the same as listed for the daily dose comparison in the legend of Figure 2.

**Figure 5 cancers-15-01377-f005:**
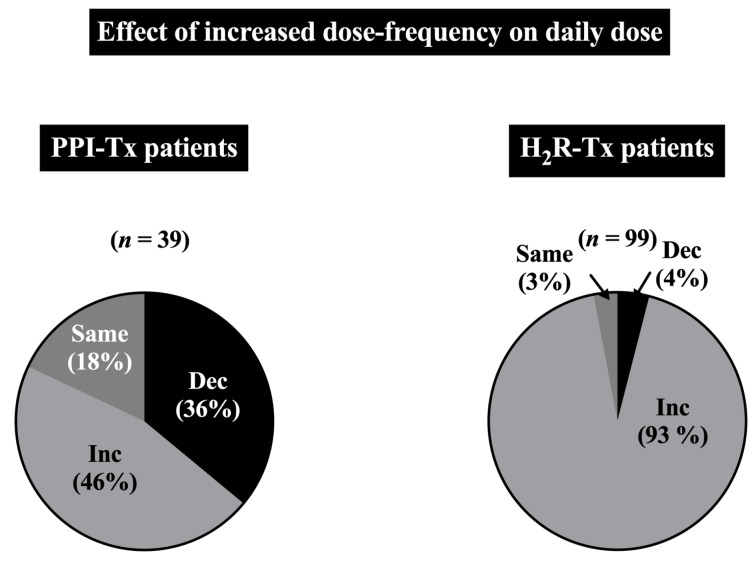
Distribution of the first and final total daily drug dosages in 303 ZES patients with long-term treatment with either a histamine H_2_-receptor antagonist or proton pump inhibitor. In the top panel, the distribution of the first and final total daily drug doses are separated for patients taking PPIs once, twice, or three times per day. In the bottom panel, the first and last total daily doses of H_2_Rs are reported. Distributions at different drug doses are reported as the percentage of the total number of patients taking the given drug class at the listed total daily dosage. In the top panel, the drug dosages are reported as omeprazole-equivalent doses for the different PPIs, determined as described in the Section 2. In the bottom panel, the total daily H_2_R doses are reported as ranitidine-equivalent daily doses, determined as described in the Section 2 and previously in ref. [109,112,178]. The mean times between doses for both classes of drugs are shown above in the legend of Figure 2.

**Table 4 cancers-15-01377-t004:** Acid treatment: comparison of initial and final daily H_2_R and PPI drug doses.

Characteristic	Number (%)		
	H_2_R (a,b)	PPI (a,c)	*p*-Value
I. Dosing (a)			
Overall (*n* = 303)			
Initial daily dose (mg/day)			
# of patients	241	262	
Mean ± SEM	968 ± 62	71.7 ± 2.3	
(Range)	(80–4800)	(20–240)	
Final daily dose (mg/day)			
Mean ± SEM	2440 ± 141	58.5 ± 2.4	
(Range)	(233–14400)	(20–240)	
% Patients with change in daily dose			
Increase	70.0%	12.6%	<0.0001
Decrease	2.1%	46.9%	<0.0001
No change	27.9%	40.5%	0.0037
% of patients with different daily doses			
Lowest initial dose (i.e., H_2_-R ≤ 900 mg/day/PPI 20 QD/20 BID)	154/229 (67%)	143/261 (22%)	<0.0001
Lowest final dose (i.e., H_2_-R ≤ 900 mg/day/PPI 20 QD/20 BID)	61/229 (27%)	128/261 (49%)	<0.0001
Higher initial doses (i.e., ≥2300 mg/day/PPI-80 mg/day)	31/229 (13%)	125/261 (49%)	<0.0001
Higher final doses (i.e., ≥2300 mg/day/PPI-80 mg/day)	138/229 (60%)	99/261 (38%)	<0.0001
Short-term acid treatment (<5 yrs) with any drug (c)			
Initial daily dose (mg/day)			
# of patients	124	60	
Mean ± SEM	1112 ± 96	63.3 ± 3.8	
(Range)	(80–4800)	(20–120)	
Final daily dose (mg/day)			
Mean ± SEM	2534 ± 204	62.7 ± 5.0	
(Range)	(90–4800)	(20–200)	
Long-term acid treatment (≥5 yrs) with any drug (d)			
Initial daily dose (mg/day)			
# of patients	114	202	
Mean ± SEM	812 ± 75	74.2 ± 2.7	
(Range)	(100–4800)	(20–240)	
Final daily dose (mg/day)			
Mean ± SEM	2341 ± 196	57.3 ± 2.7	
(Range)	(200–12000)	(20–240)	

(a). Both the H_2_R and PPI control drug doses were established by determining the minimum drug dose that reduced the gastric acid output in mEq/h for the hour prior to the next drug dosage to the endpoint, as previously described [37,59,108,112,169,201] and in the Section 2. (b). The total daily H_2_R doses are reported as ranitidine-equivalent daily doses, determined as previously described [109,112] and in the Section 2. (c). PPI drug dosages are reported as omeprazole-equivalent doses, as described in the Section 2. (d). For explanation of short and long-term treatment, see Table 1 footnote (a).

**Table 5 cancers-15-01377-t005:** Acid treatment: comparison of initial and final daily H_2_R and PPI dosing frequencies.

Characteristic	Number (%)		
	H_2_-R (a,b)	PPI (a,c)	*p*-Value
I. Dosing (a)			
Overall (*n* = 303)			
Initial daily dosing frequency			
# of patients	241	262	
Mean ± SEM	2.85 ± 0.07	1.47 ± 02.3	<0.0001
(Range)	(1–6)	(1–3)	
Final daily dosing frequency			
Mean ± SEM	3.53 ± 0.07	1.48 ± 0.3	<0.0001
(Range)	(1–6)	(0–3)	
% Patients with change in daily dosing frequency			
Increase	99/229 (43%)	39/262 (15%)	<0.0001
Decrease	13/229 (5.6%)	33/262 (13%)	<0.0001
No change	117/229 (51%)	190/262 (72%)	0.0037
% of patients with different daily doses			
Lower initial frequency (i.e., H_2_-R ≤ 3×/day/PPI ≤ 1×/day)	104/241 (43%)	142/262 (54%)	<0.0001
Lower final frequency (i.e., H_2_-R ≤ 3×/day/PPI ≤ 1×/day)	44/229 (19%)	134/262 (51%)	<0.0001
Highest initial frequency (i.e., ≥4×/DAY-H_2_-R/ day/ > 2/day PPI)	86/241 (36%)	2/262 (0.76%)	<0.0001
Higher final frequency (i.e., ≥4×/H_2_-R/ day/ > 2/day PPI)	150/229(66%)	2/262 (0.76%)	<0.0001
Short-term acid treatment (<5 yrs) with either H2R or PPI (c)			
Initial daily dosing frequency			
# of patients	126	60	
Mean ± SEM	2.76 ± 0.10	1.28 ± 0.6	
(Range)	(1–6)	(1–2)	
Final daily dosing frequency			
Mean ± SEM	3.54 ± 010	1.32 ± 0.07	
(Range)	(1–6)	(1–2)	
Long-term acid treatment (≥5 yrs) with either H2R or PPI (c), (d)			
Initial daily dosing frequency			
# of patients	117	202	
Mean ± SEM	2.93 ± 0.09	1.52 ± 0.4	
(Range)	(1–5)	(1–3)	
Final daily dosing frequency			
Mean ± SEM	3.52 ± 0.10	1.52 ± 0.4	
(Range)	(1–6)	(1–2)	

(a). Both the H_2_R and PPI control drug doses were established by determining the minimum drug dose that reduced the gastric acid output in mEq/h for the hour prior to the next drug dosage to the endpoints previously described [37,59,108,169,175,201] and in the Section 2. (b). The total daily H_2_R doses are reported as ranitidine-equivalent daily doses, determined as previously described [109,112] and in the Section 2. (c). PPI drug dosages are reported as omeprazole-equivalent doses for the different PPIs, determined as described in the Section 2. (d). For an explanation of short and long-term treatment, see Table 1 footnote (a).

**Table 6 cancers-15-01377-t006:** Treatment outcomes-acid/symptoms.

Characteristic	Number (%)		
	H_2_R (a,b) (*n* = 245)	PPI (a,b) (*n* = 262)	*p*-Value
I. Acid control on treatment (a)			
A. All patients- BAO -no drug (mEq/h)			
Mean ± SEM	40.1 ± 1.54	42.2 ± 1.6	
(Range)	(1.8–159)	(6–159)	
B. All patients- H_2_R/PPI control (a) (mEq/h)			
Mean ± SEM	4.16 ± 0.46	2.16 ± 0.16	<0.0001
(Range)	(01–10.5)	(0–9.8)	
Final acid control (mEq/h) (a,b)			
Mean ± SEM	3.31 ± 0.40	0.97 ± 0.12	<0.0001
(Range)	(0–10.4)	(0–7)	
C. Level of acid control in all patients- H_2_R/PPI (a) (% total at initial/final treatment)			
<1 mEq/h (c)	39/130 (30%)	207/377 (55%)	<0.0001
1–5 mEq/h (c)	53/130 (22%)	138/377 (37%	0.40
≥8–10 mEq/h (c)	23/130 (14%)	11/377 (2.9%)	<0.0001
D. Outcome of acid control-all patients			
Control symptoms/acid-peptic mucosal disease	100%	100%	
Not control symptoms/acid acid-peptic mucosal disease	0%	0%	

(a). Both the H_2_R and PPI control drug doses were established by determining the minimum drug dose that reduced the gastric acid output in mEq/h for the hour prior to the next drug dose to the endpoint, as previously described [37,59,108,169] and in the Section 2. (b). Included all patients treated with either H_2_R or PPI at any time. (c). For the <1, 1–5, and 8–10 mEq/h acid control groups for both the H_2_R- and PPI-treated patients, the results are expressed as the number of patients with acid control levels within this range for either initial or final acid control over the total number of patients that were included in this assessment.

**Table 8 cancers-15-01377-t008:** Summary of key findings.

Nr.	Summary of Key Findings
1.	Long-term control (mean 14 yrs)/lifelong control (30%) as well as acute control of gastric acid hypersecretion by antisecretory medications was successful in all in 303 ZES patients.
2.	This was only possible by individually regulating antisecretory drug doses using proven acid inhibitory secretory criteria and adjusting doses accordingly, combined with assessments of acid-peptic/GERD symptoms and UGI endoscopic findings.
3.	Both H_2_Rs and PPIs were successfully used for both acute and maintenance control of acid hypersecretion; however, because of their greater potency and longer duration of action, PPIs are now routinely used in almost all patients.
4.	During the initial drug dosing, the mean PPI dose was 72 mg/day, whereas with long-term treatment, half of ZES patients could have their daily PPI total dose reduced and 13% required an increase in their daily PPI dose. In contrast, with H_2_Rs, the mean initial dose was 969 mg/day of a ranitidine-equivalent dose (RED), with 70% of patients requiring increased daily doses and only 2% tolerating a total H_2_R daily dose decrease, resulting in an almost 3-fold increase in final mean final H_2_R daily dose (i.e., 2440 mg RED/day).
5.	During long-term control of acid hypersecretion, total PPI daily use could be reduced to the lowest dose level of 20 mg QD/20 mg BID, with a 2-fold higher percentage of patients than during initial treatment, whereas the opposite occurred with H_2_Rs, with a 2.6-fold lower percentage of patients requiring the lowest dose.
6.	H_2_Rs also differed from PPIs in dosing frequency. At the initial acid control, the mean frequency of H_2_Rs was almost 3 times/day, whereas it was 1.5 times/day for PPIs. On the final dose, the frequency of H_2_Rs had increased further to 3.5/day and the PPI mean frequency remained unchanged. These changes were amplified by individual patient drug frequency changes, which showed an average 3-fold greater increased dose frequency required for H_2_Rs compared to no change with PPIs.
7.	The use of PPIs at both the initial and final drug dosing demonstrated markedly better (*p* < 0.001) acid control than with H_2_Rs.
8.	Some clinical (i.e., severe GERD), laboratory (BAO, MAO), and initial acid treatment variables (presence of achlorhydria, PPI dosing frequency) correlated with the need for subsequent PPI dose changes, raising the possibility of selectively identifying patients needing dose changes and tailoring acid testing based on these variables.

## Data Availability

The data presented in this study are available in this article.

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
