# Peer review of "Successful Lifetime/Long-Term Medical Treatment of Acid Hypersecretion in Zollinger-Ellison Syndrome (ZES): Myth or Fact? Insights from an Analysis of Results of NIH Long-Term Prospective Studies of ZES"

_cancers, 2023, doi:10.3390/cancers15051377_

Round 1

Reviewer 1 Report

This is an excellent study evaluating the longitudinal management of ZES by investigators with a large cohort of patients with ZES. The results of this large study are well presented in the manuscript. The role of acid suppressive medications is well described. Given the large experience with PPI therapy it would be useful to include potential adverse events associated with the high dose PPI therapy. For example, was there an increased incidence of dementia, cardiovascular disease or hip fracture?

Author Response

  1. As the reviewer suggests we would like to include data on our patients of the occurrence of possible drug related side-effects reported in the literature from mainly epidemiological studies. Unfortunately, we do not have that data available and are only starting to try to extract it. As we mention in the simple summary on page 1  when we extracted the data for this study, which took longer than 3 months , we only concentrated on drug dose and efficacy because we thought that the side-effects from other studies such as chronic GERD were also applicable to our patients, whereas no efficacy data existed. The extraction of the possible side-effects will likely take even longer because this data requires review of numerous other data sources for our patients such as results from clinical chemistry, hematology, and radiological databases as well as a review of detailed progress notes. As I mentioned above, we are now starting to do this. We have already reported the short-term results with PPIs decreasing vitamin B12 (ref. 176, PMID#9626024. Am J Med. 1998 104, 422-30)  and not effecting body iron stores in our patients (ref 229,Stewart C, {PMID#9692706, Alim Pharm. Therap. 1998:12; 83-98). We hope to include additional data on this in the long-term data for all side-effects we are now working in the   What is clear from our data is no patient on PPIs/2nd-3rd generation H2Rs was withdrawn because of a side-effects. In our early studies in the 1980s, when initially only cimetidine was available some patients treated with very high dose of cimetidine developed antiandrogen-side-effects, (impotence, gynecomastia) and were switched to ranitidine as soon as we got it. This has also been previously reported (ref. 171).We have added this information to the paper on page 6 at the bottom of the 2nd paragraph in yellow in the marked copy  

Reviewer 2 Report

I’m very glad to review the paper in greater depth because the subject is interesting. This is a well-written paper containing interesting results which merit publication. For the benefit of the reader, however, a number of points need clarifying and certain statements require further justification. There are given below:

1.       The paper includes patients treated for short-term (<5 years) and long-term (>5 years) periods. Why choose “5-year” as dividing point? Please make explanation or provide literature support.

2.       There are two therapeutic goals in patients with ZES: to control the hypersecretion of gastric acid and to treat the gastrinoma itself. Acid hyper-secretion can be controlled by medical treatment, but surgical treatment is the only way to eradication of the tumor and cure the disease. Comparisons between medical and surgical treatments should be made in the “Discussion” section.

3.       There are some spelling mistakes and style errors in the text. Thus, your manuscript needs careful editing by someone with expertise in technical English editing so that the goals and results of the study are clear to the reader.

Author Response

  1. The division of the patients into a short-term and long-term group based on treatment at NIH for above or below 5 year was performed because in different studies in the literature, almost all ZES patients previously reported had only been treated with acid antisecretory agents for <5 years. We wanted to compare our patients to these, while we know there would be no comparative groups for those in the long-term treatment group. We also wanted to divide the patients into these two groups because we thought the length of follow-up could affect the results, which we found was not the case. We have added these point to the revised paper on page 5 in yellow in the marked copy.
  2. We completely agree with the reviewer. We have added a section to the discussion as recommended emphasizing the importance of both approaches to the tumor through surgery, etc and to the medical control of the acid hypersecretion. Please see the final paragraph of the discussion in yellow in the marked copy.
  3. We have had two authors who are native English speakers review the paper carefully to correct any grammar, spelling or unclear syntax.